# Qualitative study of practices and attitudes towards physical activity among prediabetic men and women in urban and rural Malawi

Jethro Banda  ,[1] Christopher Bunn,[1,2,3] Amelia C Crampin,[1,3,4] Jason M R Gill,[5] Cindy M Gray  [2,3]

¹Malawi Epidemiology and Intervention Research Unit, Lilongwe, Malawi
²School of Social and Political Sciences, University of Glasgow, Glasgow, UK
³School of Health and Wellbeing, University of Glasgow, Glasgow, UK
⁴London School of Hygiene and Tropical Medicine, London, UK
⁵School of Cardiovascular and Metabolic Health, University of Glasgow, Glasgow, UK

**Correspondence to**
Professor Cindy M Gray;
cindy.gray@glasgow.ac.uk

## ABSTRACT

**Objectives** Given the decline in physical activity levels in Malawi, like other sub-Saharan African countries, and its implication for non-communicable disease (NCD) prevention, this study aimed to compare and contrast accounts of practices and attitudes towards physical activity among Malawian men and women (previously identified as having pre-diabetes) in urban and rural settings.

**Setting** Two communities: one urban (Lilongwe) and one rural (Karonga).

**Participants** 14 men (urban N=6, rural N=8) and 18 women (urban N=9, rural N=9) classified as prediabetic during their participation in an NCD survey 3-5 years previously.

**Design** A qualitative focus group study (N=4) and thematic analysis, with the ecological model used as a framework to characterise the types of physical activity people engaged in and potential ways to support them to exercise more.

**Results** Participants reported undertaking different types of physical activity across all ecological model domains (household, occupational, transport, recreational). Rural participants reported more vigorous physical activities than urban participants, and women reported more household activities than men. Many participants recognised a need to promote physical activity in Malawi, and the health benefits of doing so, including the importance of physical activity in helping them stay strong to maintain physical functioning. Barriers to physical activity included competing priorities (especially urban men), societal expectations around wealth, use of motorised transport, lack of accessible facilities for women, ageing and ill health.

**Conclusions** Physical activity is declining in Malawi as working and transport practices change in response to economic development, making promotion of alternative forms of physical activity a public health priority. Multilevel interventions emphasising the personal benefits/value of physical activity for all ages, and routine and group-based exercising, as well as investment in accessible recreational facilities (including for women) and active travel infrastructure should be considered to improve physical activity levels in Malawi.

## STRENGTHS AND LIMITATIONS OF THIS STUDY

⇒ Conducting separate focus groups with men and women in urban and rural settings allowed systematic comparison of urban/rural and gender differences in physical activity.
⇒ Participants had previously been told they were at high risk of developing diabetes and may been more informed about the importance of physical activity for their health than the general adult population of Malawi.
⇒ While focus groups enabled participants to build on each other's accounts, no examination of the extent to which the group context influenced individual responses (eg, conformity) was undertaken.

mortality globally.[1] The WHO recommends that adults undertake 150 min of moderate-intensity or 75 min of vigorous-intensity physical activity a week for optimal health.[2] However, a quarter of adults fail to meet these guidelines worldwide.[2] In sub-Saharan Africa, physical activity levels have historically been high, but increased urbanisation and adoption of Westernised lifestyles mean physical inactivity is increasing.[3–6] In Malawi, the most recent WHO country profiles published in 2011 and 2018 demonstrate that the prevalence of physical inactivity rose from 10% to 14% in the 8 years to 2016.[7 8] In addition, some studies have reported lower physical activity levels in Malawi's urban areas,[9] with one population survey reporting that 24% of urban residents were inactive compared with 9% of rural residents.[10] Gender differences are less clear: while population surveys report that men in Malawi are generally more physically active than women[10 11]; some studies in specific urban and rural settings show that women are more active than men.[12 13]

The prevalence of non-communicable diseases (NCDs), such as diabetes, cardiovascular disease, chronic respiratory disease and

## INTRODUCTION

Lack of physical activity is thought to be the fourth leading risk factor for premature

cancer, is on the rise in sub-Saharan Africa.[14 15] Deaths from NCDs in low- and middle-income countries are forecast to increase from 31.5 million in 2016 to 41.9 million in 2030, and to 56.6 million by 2045.[16] Recent estimates suggest that 32% of deaths in Malawi are associated with NCDs.[1] Although being physically active helps both prevention and management of NCDs,[17–19] currently Malawi has no physical activity public health campaigns.[1]

Promotion of behaviour change for physical activity requires detailed understanding of current local practices and attitudes, as well as the different contexts in which physical activity is undertaken. However, only a handful of qualitative studies exploring practices and attitudes towards physical activity in sub-Saharan Africa have been conducted. Interviews and focus group discussions with urban women in Ghana revealed a desire for more social (including group-based) physical activity, and that lack of time and concerns about injury (eg, doing age-appropriate exercises) were barriers to being physically active.[20] Three studies with women in urban South Africa[21–23] showed that although participants were aware of the physical and mental health benefits of physical activity (including weight loss), lack of time, money and information, perceptions of local crime, and negative influences from other people, including abuse from men for wearing 'exercise' clothing, were barriers to their being more physically active. Focus groups with men and women with suspected type 2 diabetes, suspected pre-diabetes or who were obese in urban and rural Uganda suggested that people were willing to become more active to improve their health, and some described doing this by embedding physical activity within their normal routines.[24] This included washing, cooking, walking and cycling in urban areas, and more vigorous activities such as gardening, bush clearing, fetching water, splitting firewood and pounding food in rural areas. However, gender differences were not explored.

Two qualitative studies on physical activity have been conducted in Malawi with men and women with diabetes. In one study, interviews and focus groups with diabetes patients attending health education classes at a rural private hospital revealed that people were aware of the importance of both routine and leisure (recreational) physical activity for their health; however, no gender comparisons were reported.[25] In a separate study, focus group discussions with patients in urban and semiurban areas reported both positive and negative social influences. Specifically, while ridicule from other people hindered their participation in some types of physical activity, encouragement from health workers and family members helped them remain active.[26]

Despite declining physical activity levels[6 15] and the implication of this for NCD prevention and management,[17–19] there is currently very little evidence about the practices and attitudes of adults in Malawi towards physical activity. As physical activity (and its determinants) may differ according to gender and place of residence,[2 27–30] the aim of this qualitative study was to compare and contrast accounts of practices and attitudes towards physical activity among Malawian men and women in urban and rural settings who had previously been identified as having pre-diabetes. This new evidence will be useful in informing the development of effective physical activity interventions in Malawi.

## METHODS

### Study design

To elicit accounts of physical activity in everyday life among urban and rural Malawians living with pre-diabetes, we used a focus group-based research design. Focus groups were chosen to enable participants to develop rich accounts of physical activity in their communities in an interactive setting that encouraged them to elaborate and examine the narratives offered by fellow participants.[31] This approach was framed within a pragmatist philosophy,[32] which positions participants' narratives as neither objects of knowledge that enable prediction nor as the 'truth' of the participants. Rather, the narratives are seen as offering insight into the 'vocabularies' of physical activity that are used by participants, and which can be engaged with when seeking change.

### Setting and recruitment

Data collection took place in Lilongwe (urban) and Karonga (rural) in March 2019. Participants were recruited from a sample of adults previously identified (during participation in an NCD survey between 2013 and 2016)[12] as having pre-diabetes. They were part of a larger Medical Research Council (MRC)-funded study (MR/R019428/1; NHSRC protocol 18/01/1951) investigating links between nutritional intake and physical, metabolic and inflammatory phenotypes in prediabetic (n=50) and normoglycaemic (n=50) men and women. During the informed consent process for this larger study, participants were asked if they would be willing to participate in a focus group if selected. No refusals were recorded during this process, and participants from the 'prediabetic' subgroup were purposively sampled to achieve variation in age.

### Participants

Four focus groups were held with men and women separately in the urban and rural locations. They included a total of 32 participants (urban N=15, 6 men, 9 women; rural N=17, 8 men, 9 women). At the time of the current study, some participants reported progression from their prediabetic status at the last NCD survey to diabetes and/or hypertension.

### Participant and public involvement

At the start of the MRC-funded study, stakeholder engagement meetings with community leaders, community members, healthcare professionals, policy makers and patient groups were held in both rural and urban locations to raise awareness of the research, and gather

feedback on the design and implementation of data collection procedures, including the focus group discussions reported here.

## Data collection and management

The focus groups lasted between 90 and 120 min and were audio recorded with participant consent. A semistructured topic guide (see online supplemental material 1) was developed informed by our pragmatist approach[32] to focus discussion around participants' personal accounts of physical activity in their daily lives, their understandings of physical activity and health (including the benefits of being physically active), perceived barriers and facilitators to being physically active (both personal and among others in their community) and their perceptions of changes in physical activity over time. The topic guide was developed in English by CB, CMG and Hazel Namadingo (HN), all experienced qualitative researchers, then translated into Chichewa for urban focus groups and into Chitumbuka for rural focus groups by fluent Chichewa and Chitumbuka speakers. The urban focus groups were conducted by HN; the rural focus groups were conducted by Cecelia Nyirenda, an experienced field worker. The facilitators were fluent in the local languages and familiar with the issues under discussion.

The audio recordings, together with field notes taken during the focus groups, were transcribed directly into English by experienced field workers fluent in Chichewa and Chitumbuka. HN (native Chichewa speaker) and JB (native Chichewa speaker and fluent in Chitumbuka) then quality assured the transcripts while listening to the audio recordings, and made changes where necessary.

## Data analysis

The quality-assured transcripts were analysed using a thematic codebook approach.[33] JB, CMG and CB read through the transcripts line by line to familiarise themselves with the data and identify initial codes relating to participants' accounts, knowledge, barriers and facilitators of physical activity, and any emerging areas of interest (such as changes in physical activity over time in Malawi). The three authors then met to discuss and combine their initial codes into broad semantic themes to reflect participants' accounts of physical activity. JB applied the broad themes to all transcripts using NVivo V.11 software to organise the data, and an iterative approach to identify all data within each theme and consider whether the themes adequately reflected the meanings represented in the data or needed refining (eg, by coding additional potentially relevant extracts for inclusion in the analysis as 'other'). The resultant coded transcripts were reviewed by CMG, any disagreements resolved through discussion, and all extracts coded as 'Other' jointly examined to decide whether they represented a new broad theme, belonged to an existing theme or could be discarded. All were either assigned to an existing theme or discarded. JB then further refined the broad themes into subthemes

to produce a detailed narrative account of each broad theme, which included careful reading, rereading and detailed notetaking using an adapted one sheet of paper (OSOP) approach[34] to compare similarities and differences in accounts between men and women in urban and rural settings, and to capture interactions within the focus groups. The ecological model of physical activity[35 36] was then applied as a framework to characterise the different types of physical activity participants engaged in. Again, an adapted OSOP approach[34] was used to compare men's/women's and urban/rural accounts. The findings presented below are illustrated by extracts with anonymised participant identifiers that include gender, age range (30-39, 40-49, 50-59, 60-69), setting (urban/rural) and any self-reported health conditions at the time of the current study.

## RESULTS

As table 1 shows, 50% (16/32) of participants were aged 50 and above. On average, urban participants were slightly younger (43.1 years, SD=12.3) than their rural counterparts (51.6 years, SD=16.8), and length of stay was much shorter in the urban (17.9 years, SD=7.2) than the rural (36.2 years, SD=17.9) setting. Overall, 53% (17/32) had not been educated beyond primary school, but urban participants were better educated than rural participants (73%, 11/15 of urban participants had secondary school or higher education compared with only 24%, 4/17 of rural participants). Most urban participants (80%, 12/15) were either self-employed or salaried, whereas almost all rural participants (94%, 16/17) were involved in subsistence activities. Over half (66%, 21/32) were married, 25% (8/32, all women) reported being divorced/widowed and 9% (3/32, all men) had never married. The participants had all been identified as having pre-diabetes during the 2013-16 NCD survey; some (9%, 3/32: one urban woman, one rural woman and one rural man) reported progression to diabetes, and 34.4% (11/32: two of whom also reported diabetes) reported a subsequent hypertension diagnosis.

Our analysis identified 10 broad themes reflecting participants' accounts of physical activity in their daily lives. Two of these described the context of physical activity in Malawi: (1) Types of physical activity undertaken; (2) Changes in physical activity over time. Six described factors that participants perceived as influencing physical activity: (3) Physical activity and health; (4) Motivations for physical activity; (5) Barriers to physical activity; (6) Technology and physical activity; (7) Social norms; (8) Social support. One described potential solutions: (9) Ways of improving physical activity. (A final theme: 10) Sources of information, is not elaborated in this paper). A detailed description of the nine themes reported here, together with example extracts, is provided in online supplemental table S1. The following sections present participants' accounts of physical activity in their daily lives. In

**Table 1** Focus group participant characteristics

| | All (N=32) (%) | Urban (N=15, 46.9%) | | | Rural (N=17, 53.1%) | | |
|---|---|---|---|---|---|---|---|
| | | Total | Female | Male | Total | Female | Male |
| **Sex** | | | | | | | |
| Male | 14 (43.8) | 6 (40.0) | – | 6 (100) | 8 (47.1) | – | 8 (100) |
| Female | 18 (56.2) | 9 (60.0) | 9 (100) | – | 9 (52.9) | 9 (100) | – |
| **Age** | | | | | | | |
| Age (years)* | 47.6 (15.3) | 43.1 (12.3) | 46.7 (11.9) | 37.7 (11.7) | 51.6 (16.8) | 54.4 (10.9) | 48.5 (22.1) |
| ≤35 | 8 (25.0) | 5 (33.3) | 3 (33.3) | 2 (33.3) | 3 (17.7) | 0 (0.0) | 3 (37.5) |
| 36–49 | 8 (25.0) | 4 (26.7) | 2 (22.2) | 2 (33.3) | 4 (23.5) | 3 (33.3) | 1 (12.5) |
| ≥50 | 16 (50.0) | 6 (40.0) | 4 (44.5) | 2 (33.3) | 10 (58.8) | 6 (66.7) | 4 (50.0) |
| **Length of stay** | | | | | | | |
| Length of stay (years)* | 27.7 (16.6) | 17.9 (7.2) | 20.1 (8.4) | 14.7 (3.0) | 36.2 (17.9) | 29.8 (12.6) | 43.5 (21.1) |
| 6–10 years | 3 (9.4) | 2 (13.3) | 2 (22.2) | 0 (0.0) | 1 (5.9) | 1 (11.1) | 0 (0.0) |
| 11–15 years | 5 (15.6) | 5 (33.3) | 1 (11.1) | 4 (66.7) | 0 (0.0) | 0 (0.0) | 0 (0.0) |
| 16–20 years | 5 (15.6) | 3 (20.0) | 1 (11.1) | 2 (33.3) | 2 (11.8) | 2 (22.2) | 0 (0.0) |
| ≥20 years | 19 (59.4) | 5 (33.3) | 5 (55.6) | 0 (0.0) | 14 (82.3) | 8 (88.9) | 6 (75.0) |
| **Highest education** | | | | | | | |
| Primary school | 17 (53.1) | 4 (26.7) | 4 (44.4) | 0 (0.0) | 13 (76.5) | 8 (88.9) | 5 (62.5) |
| Secondary and higher | 15 (46.9) | 11 (73.3) | 5 (55.6) | 6 (100) | 4 (23.5) | 1 (11.1) | 3 (37.5) |
| **Employment** | | | | | | | |
| Not working | 2 (6.3) | 2 (13.3) | 2 (22.2) | 0 (0.0) | 0 (0.0) | 0 (0.0) | 0 (0.0) |
| Subsistence | 17 (53.1) | 1 (6.7) | 1 (11.1) | 0 (0.0) | 16 (94.1) | 8 (88.9) | 8 (100) |
| Self-employed | 8 (25.0) | 8 (53.3) | 5 (55.6) | 3 (50.0) | 0 (0.0) | 0 (0.0) | 0 (0.0) |
| Salaried | 5 (15.6) | 4 (26.7) | 1 (11.1) | 3 (50.0) | 1 (5.9) | 1 (11.1) | 0 (0.0) |
| **Marital status** | | | | | | | |
| Never married | 3 (9.4) | 1 (6.7) | 0 (0.0) | 1 (16.7) | 2 (11.8) | 0 (0.0) | 2 (25.0) |
| Married | 21 (65.6) | 11 (73.3) | 6 (66.7) | 5 (83.3) | 10 (58.8) | 4 (44.4) | 6 (75.0) |
| Widowed/divorced | 8 (25.0) | 3 (20.0) | 3 (33.3) | 0 (0.0) | 5 (29.4) | 5 (55.6) | 0 (0.0) |
| **Reported health condition** | | | | | | | |
| Prediabetic only | 20 (62.5) | 11 (73.3) | 5 (55.6) | 6 (100) | 9 (52.9) | 5 (55.6) | 4 (50.0) |
| Diabetic only | 1 (3.1) | 0 (0.0) | 0 (0.0) | 0 (0.0) | 1 (5.9) | 1 (11.1) | 0 (0.0) |
| Hypertensive only | 9 (28.1) | 3 (20.0) | 3 (33.3) | 0 (0.0) | 6 (35.3) | 3 (33.3) | 3 (37.5) |
| Diabetic and hypertensive | 2 (6.3) | 1 (6.7) | 1 (11.1) | 0 (0.0) | 1 (5.9) | 0 (0.0) | 1 (12.5) |

Data are presented as n(%), or *mean(SD)

particular, they focus on the context in relation to the types of physical activity people described doing and how these have changed over time, the factors influencing physical activity and potential ways to improve physical activity in Malawi.

### What do adults in Malawi do when they are physically active?

The ecological model constructs physical activity as influenced by interacting factors operating at different levels: individual, interpersonal, environmental (social, built and natural) and policy.[35] It further organises physical activity around four active living domains: household, occupational, transport and recreational.[36] Our analysis of the types of physical activity undertaken showed that Malawian men and women in both urban and rural settings were active across all four ecological model domains (see also online supplemental table S2). These are now elaborated in turn.

### Household physical activity

Overall, women reported doing more household physical activity than men. In both urban and rural settings, women appeared to take the lead in housework and preparing meals:

> As a woman you are supposed to do physical activities which are household chores. We sweep around the surrounding, we prepare breakfast, we mop the house, then we go to the market to buy relish and some other things. When you come back, time is already up for lunch, you start preparing lunch… (Female, 50-59, urban, hypertension reported)

Rural women also described carrying wood, collecting food from their farms for meals and pounding maize or cassava for flour. Likewise, rural men reported

undertaking vigorous household activities, including digging and cutting wood:

> As for me, I am physically active in the morning, such as cutting wood, weeding [gardening] at home […] In the afternoon, I rest. And thereafter, around 4 pm, I start again clearing outside the house… (Male, 70-79, rural, hypertension reported)

By comparison, urban men described less strenuous household activities, such as car washing and maintenance. Some also described involvement in household chores and cooking:

> I sweep the surrounding, I mop, I cook.
>
> Facilitator: Do you stay alone?
>
> No, I live with my relatives, but I help them [with] household chores. (Male, 40-49, urban, no health conditions reported)

### Occupational physical activity

For rural participants, most work appeared to involve effortful physical activity. Both women and men reported daily involvement in farming activities, including planting, weeding and harvesting their crops. One woman said:

> A person has a lot of work to do, like in this rainy season, we do farming. Instead of sleeping, you think of going to the garden to do farming activities […]. In the afternoon, if you left some ridges which you didn't plant the seeds, you go back and plant the stems. (Female, 50-59, rural, no health condition reported)

As this extract indicates, there was some overlap between household and occupational physical activity in the accounts of rural participants. Similarly, many rural men described fishing on Lake Malawi, both to provide food for their families and to sell to make money. (Like farming, fishing involves vigorous physical activity, including launching and rowing boats, and landing catches.) In contrast, urban men talked about running shops and market stalls, working in the service industry and having office jobs, which, as one man described, were often largely sedentary:

> Mostly I do my work while seated working on the computer. If I move around, that means I'm just delivering files to my boss's office. When I knock off from work, I use a car… (Male, 50-59, urban, no health condition reported)

Many urban women did not work outside the home. The exceptions were two women who had their own businesses. One of these described how running her market stall involved heavy physical work:

> …I sell timber at [local] market. I cut the timber myself, I bought an electric saw blade. I have a wheelbarrow, I use it to carry sand when the rains fall. I push it

myself as part of exercise. (Female, 40-49, urban, no health condition reported)

### Transport-related physical activity

Most rural participants reported that walking was still their main mode of getting from place to place. Many described walking to work (to their farm or to go fishing), to the market and to visit friends or family. Urban participants also reported walking as part of their daily routines, but to a lesser extent than their rural counterparts. Many preferred to use public transport, taxis (including bicycle taxis) or their own cars instead of walking:

> I cannot go to some places by foot while I have a car. I would prefer to fill my car with fuel than walking, despite that we are making our bodies not to be physically active. (Male, 50-59, urban, no health condition reported)

Importantly, despite motorised transport often being less available outside towns and cities in Malawi, some rural participants also described avoiding walking where possible:

> Nowadays it's different from the past. In the past, our parents could walk by foot from here to South Africa. But currently if I am going to Jetty [less than 5 km], I prefer to use a car. (Male, 50-59, rural, hypertension reported)

### Recreational physical activity

Both rural and urban participants reported involvement in a range of recreational physical activities, including traditional dancing (apart from urban men); jogging (apart from rural women), cycling (men only), playing with their children (mainly women, but also mentioned by men) and team sports (eg, netball for women and football for men). Football was highly popular among both urban and rural men, and its importance as a recreational activity is illustrated by this account from one younger rural man describing how he and his friends often improvise so they can enjoy a game of football after work:

> During rainy season we like playing football in a club. But in the summer season we do farming in marsh gardens, and after that we play football using a ball made from plastic papers and rubbers. (Male, 20-29, rural, no health condition reported)

A number of urban (but not rural) participants described going to the gym or attending exercise classes. As one woman highlighted, involvement in formal types of exercising helped to compensate for the largely inactive lifestyles that people living in towns and cities often led:

> … here in town we find the gym to be our solution in making our bodies active, while in the villages they work, hence making their bodies active…. (Female, 60-69, urban, hypertension reported)

### Decline in physical activity in Malawi over time

In general, people living in rural areas reported doing more strenuous physical activity within the home and at work, and also walking more than those living in the city. However, most participants felt that physical activity levels within the general population were decreasing in Malawi. They were also aware that being less active placed them at a disadvantage compared with previous generations:

> Our parents were walking long distances with heavy luggage on their heads, up to 10 km on foot. Even farming they could do by themselves without employing other people to help them. But nowadays we can't manage all these. Our bodies are weak while our parents were strong. (Female, 30-39, urban, no health condition reported)

### Factors influencing physical activity in Malawian adults

The seemingly inevitable decline in household, occupational and transport-related physical activity, suggests the promotion of physical activity (including recreational) should become a public health priority in Malawi. In order to inform the development of effective physical activity interventions, the following sections explore factors that support and prevent people in Malawi from being physically active (see also online supplemental table S3).

### Health

Participants in all focus groups agreed that being physically active was important for their general health and well-being. Some participants described how their diagnosis as being prediabetic in the 2013-16 NCD survey had acted as a catalyst for them to become more active:

> As for me, I started to do exercises regularly when I was told that I am pre-diabetic. And each time I go to the hospital, diabetes [blood sugar] is high[er] - this motivated me to do more exercises to reduce my BP and blood sugar level. (Female, 50-59, urban, hypertension reported)

Discussions within all focus groups also raised the role of physical activity in losing weight. One man directly attributed the health improvements he had experienced following his diagnosis to the weight he had lost through being active:

> Before I was diagnosed with diabetes, I was overweight, but after the physical activities - I do activities such as digging, working in rice farm, and I found that my weight has been reduced. But also sugar level has gone down, and I was told not to continue testing my sugar. So, I realize that it's because of exercises. (Male, 60-69, rural, diabetes and hypertension reported)

Others described how being physically active helped with releasing tension and maintaining flexibility.

Importantly, as the following extract illustrates, there was general agreement in all focus groups about the need to do physical activity to remain strong:

> When a person is doing physical exercise, the body gains strength if the body was weak and doesn't feel good. But after doing exercise the body feels well. (Female, 60-69, rural, no health conditions reported)
>
> Facilitator: Is there anything else on the reasons we are told to do physical exercise? […]
>
> … if you just stay idle, you don't have strength in your body (Female, 60-69, rural, hypertension reported)

### Motivations and barriers to being physically active

In addition to the consensus around health benefits, rural participants described necessity - having to provide and care for their families - as their main motivator for being physically active. As one woman explained, livelihood and physical activity are still often inextricably linked in rural communities:

> The reason we are physically active especially here on earth, it's because we want to eat. Without this, there won't be anything to eat. (Female, 50-59, rural, diabetes reported)

Family and work responsibilities also featured in the urban focus group discussions, but here their negative impact on physical activity was the dominant discourse. For example, urban men agreed the multiple competing pressures they experienced in their daily lives left little time for recreational physical activity. As one participant suggested, making money is the number one priority for many men living in towns and cities:

> Being physically active is a very good thing and it helps a lot of things in our bodies, but it seems here in town we are busy making money. Taking care of family and feeding our parents make us forget everything about ourselves, we can't have time to do exercises, which is very bad. (Male, 50-59, urban, no health condition reported)

The move from active to more sedentary occupations and lifestyles, and the pressure to work hard to make money to provide for the family, were therefore major barriers to being physically active among urban participants. Other common barriers that were mentioned in all focus groups included poor diet, age and fear of injury, which, as one rural participant explained, prevented some older women from taking part in local recreational activities:

> All these things we do at home, because we are aged and we can't participate in netball which the young ladies are doing. We can't play with them because we fear to be pulled down. (Female, 60-69, rural, hypertension reported)

In addition, the view that people who were experiencing ill-health should avoid being active was widespread, particularly, as the following extract illustrates, when illness was combined with age:

> I was leaving everything for my father to do, while I was just staying. But after some time he was diagnosed with high blood pressure and other diseases. So I said to my father that he should not work anymore, but I should be the one doing physical work while he should be resting. (Male, 30-39, rural, no health condition reported)

However, some participants admitted they could not always blame external factors for their inactivity. Many talked about 'laziness' and 'developing laziness' as their only excuse:

> Facilitator: What do you think would stop you from doing [exercises] regularly?
>
> Laziness. (Female, 50-59, urban, no health conditions reported)
>
> Same… laziness (Female, 40-49, urban, no health conditions reported)

Interestingly, this tendency to describe oneself as 'lazy', which appeared to have negative connotations and to be associated with risk of disease and weakness, was evident across all focus groups, even among rural men and women whose daily physical activity levels were often extremely high.

### Technology

The negative influence of technological advancements on physical activity emerged as an independent theme in our analysis. With the exception of the increased availability of motorised transport (which is discussed further in the next section), technology mainly seemed to be having an impact in rural areas. Rural men described how new tools and machinery had taken some of the hard work out of farming, and rural women reported how being able to take their maize for processing at the local mill, instead of doing it themselves at home, was a symbol of their social standing:

> …at present I can take 50 kgs of maize and start pounding, but people will get surprised of me - why I am doing such activity while am having maize mill closer to me? So instead of doing such activity, I try hard to get money and take the maize to the maize mill. (Female, 50-59, rural, diabetes reported)

### Social norms

The societal value of wealth generation was discussed in all focus groups, and, as the previous extract suggests, having money in Malawi is associated with social expectations that further limit people's opportunities for physical activity. For example, those who can afford it, tend to employ other people to do the physical work (eg, household chores and farming) that they and their family previously had to do for themselves. This trend was particularly evident in the urban discussions:

> … maybe the only difference [from people living in rural areas] is farming, they use their own hands… we living here in town, we will search for piece-work [casual work] men to do that for us and then pay them later, that's why our life seems to be weak. (Female, 30-39, urban, no health condition reported)

It was also evident that walking from place to place is not something that Malawians who can afford other forms of transport do through choice. One urban man clearly described how, as he earned more money, the amount of physical activity he expended in getting to work gradually declined:

> I was walking to and from [work], then I got a promotion and I started using public transport when going to work. Because I had managed to get another promotion, I started going to work using company car and now that I have my own car, I use it as well. I have completely forgotten walking, just because I have money and everything, and this has contributed to my low level of physical activity. (Male, 50-59, urban, no health condition reported)

Expectations of what other people would think or say emerged as a strong influence on the amount and types of physical activity people did, particularly among urban participants. One man described how being made fun of because of his weight had put him off cycling:

> I take my bike to the market. People laugh at me sometimes because I am overweight. And when they see me riding a bicycle, they say I will break the bike… because they're used in seeing me in the car. What people say, it can discourage you from doing exercises. (Male, 20-29, urban, no health condition reported)

Likewise, some urban women described how gender and age-related norms limited the types of physical activity that were appropriate for women to do, and how the predominantly (younger) male clientele in many gyms made them feel uncomfortable:

> In [local gym] women cannot manage. It is full of the youths and men. (Female, 40-49, urban, no health condition reported)

Children were also reported as having a negative influence on people's (mainly women's) physical activity. Some women recounted experiences of children laughing when they engaged in recreational physical activity. Others described strong social expectations that children should help with physical household chores:

> My neighbors asked me why I draw water by myself when I have children. I tell them it's part of my exercises. (Female, 50-59, urban, no health conditions reported).

## Social support

Despite children's negative impact on some aspects of physical activity, many participants (particularly women) spoke about how their children and grandchildren helped them be active by playing sports and other (traditional) games with them:

> I invite my grandchildren to play Jingle [traditional physical game], because if I just sit down, I feel like my legs are tied up. So I jump a little bit, and my grandchildren also do the same. (Female, 60-69, rural, no health condition reported)

Similarly, involvement in sports (eg, football, netball) organised by local community groups, including churches, was seen as beneficial by participants in all focus groups. One urban woman said knowing that other people were depending on her was far more motivational than doing physical activity alone:

> I think doing on our own is what makes us fail to exercise. Whenever we feel tired, we stop the routine. But creating groups in our locations can strengthen the desire of doing exercises. If we can just be 15, we can always be meeting, and one may feel bad to be absent a single day because you feel your friends will be missing your part, if it is a team. (Female, 40-49, urban, no health condition reported)

### Suggestions to help Malawian adults increase physical activity

Our analysis shows that although other people often had a negative influence on Malawians' physical activity, social forms of recreational physical activity were clearly helpful in encouraging people to be active in both urban and rural settings. Other suggestions to promote recreational and other types of physical activity were mainly targeted at an individual level. These included providing more information about the benefits of physical activity, and about making physical activity part of normal routine (in both urban and rural focus groups), promoting good time management (particularly in urban focus groups), and, importantly, ensuring that people find activities they enjoy doing.

> What the heart has liked, it can't be stopped. Whenever you wake up you will do it. (Female, 40-49, rural, hypertension reported)

## DISCUSSION

This qualitative study demonstrates that men and women previously identified as having pre-diabetes in rural Malawi tend to engage in more vigorous physical activity in their daily lives than their urban counterparts. Recreational physical activities were practiced in both settings, but tended to focus around social interaction and having fun in the rural setting, while there was a move towards more structured forms of exercise, including the use of gym facilities, in the urban setting. Involvement in recreational physical activities reflected a recognition of the need to remain active as the amount of physical activity involved in everyday living decreases in Malawi. Indeed, the understanding that physical activity is important for maintaining good health was widespread. However, ageing, ill health and fear of injury emerged as perceived individual level barriers to being active across all focus groups, and these were compounded by lack of time due to interpersonal level barriers such as competing family and work responsibilities, wealth accumulation and negative social influences, particularly in the urban setting.

Although quantitative studies have previously demonstrated differences in physical activity between urban and rural areas of Malawi,[9 10] to our knowledge, this is the first in-depth qualitative exploration of the factors underpinning urban/rural and gender differences in physical activity among Malawian adults. It suggests that the urban/rural differences in part reflect the nature of work that people in rural areas are involved in, and necessity, with most rural occupations, such as farming and fishing, and household chores involving high levels of physical effort (although this may be changing with the advent of new technologies). In addition, the growth of motorised transportation in cities, already evident in more developed sub-Saharan Africa countries like South Africa,[37] now also appears to be negatively impacting on physical activity in urban Malawi. However, it is important to note that social expectations surrounding wealth mean that a preference to use motorised transport rather than walking, where (financially) possible, was also evident among rural participants.

Gender differences in the types of physical activity undertaken, in particular women doing more household physical activity than men, reflects the fact that the traditional gender divide (with women bearing the main responsibility for domestic and family activities) remains in Malawian society.[38] Interestingly however, the role of physical activity in maintaining strength as people age was highlighted by women as well as men. The importance of doing regular strength-based physical activity has long been recognised in WHO physical activity guidance,[39 40] and has gained prominence in the latest UK physical activity guidelines in an effort to increase adherence.[41] The value of strength-based exercises in maintaining functionality and avoiding frailty[42] is likely to be more salient (and therefore potentially motivational) in Malawi, where, as elsewhere in sub-Saharan Africa, older people often play a central role in bringing up children and running small holdings, when younger family members have moved to seek work in cities.[43]

Both men and women in the current study reported a desire to lose weight as a motivator for physical activity. This finding contradicts previous research suggesting that a larger body size is socially desirable among men and women in Malawi and other sub-Saharan African countries,[44 45] and that a lower body weight is often associated with HIV/AIDS and other diseases, stress and

**Table 2**  Strategies to improve physical activity levels in Malawi mapped against individual, interpersonal, environmental/ organisational and policy-level barriers

| Ecological model level | Barrier/opportunity | Strategy |
|---|---|---|
| Individual | Barriers<br>▶ Ill health and fear of injury<br>▶ Age | 1. Provide information on health benefits of physical activity.[2 54]<br>2. Provide information on exercising safely, including as people age.[2 53 68] |
| | Opportunities<br>▶ Maintaining health and well-being<br>▶ Losing weight<br>▶ Maintaining strength<br>▶ Willingness to exercise at home<br>▶ Recognition of importance of enjoyment | See 1 and 2.<br><br>3. Provide information on easy to do exercises, which can be done alone without support.[53 68]<br>4. Promote a range of activities (including offering taster sessions) to allow people to find something they enjoy doing.[2 69] |
| Interpersonal | Barriers<br>▶ Discouragement (including gender-based) from others<br>▶ Wealth-related societal expectations<br>▶ Family responsibilities<br>▶ Time constraints due to competing priorities | 5. Run information campaigns to normalise and promote the value of different forms of physical activity, including for both women and men.[2 54]<br>6. Encourage family involvement in physical activity to motivate each other.[54 55]<br>7. Promote forms of physical activity that can fit into daily routines (eg, walking).[53 68 69]<br><br>See also 4. |
| | Opportunities<br>▶ Playing with children<br>▶ Recognition of the value of group-based activities | 8. Promote the benefits (including bonding) associated with exercising with children.[70]<br>9. Encourage group-based physical activity.[53 68 69]<br>See also 4., 6. and 8. |
| Environmental/ organisational | Barriers<br>▶ Urban gyms dominated by (young) men<br>▶ Fewer formal physical activity facilities in rural areas | 10. Set aside female-only and other inclusive time slots in gyms.[68]<br>11. Invest in recreational infrastructure such as parks and gyms accessible by both men and women in rural, as well as urban, areas.[2 56 68]<br>12. Encourage people (including those in urban areas) to exercise outdoors using footpaths and parks.[56]<br><br>See also 5. |
| | Opportunities<br>▶ Churches, schools, community organisations<br>▶ People engage in sedentary activities at work | 13. Encourage church, school and community facilities to promote (social) physical activities.[2 68]<br>14. Promote workplace initiatives to improve physical activity (eg, walking breaks; gym membership) with incentives for participation.[2 53 54 56 57 69]<br>15. Encourage people to park a walking distance from their workplace.[53] |
| Policy | Barriers<br>▶ Growth in use of private cars | 16. Improve the quality and accessibility of public transport systems.[68]<br>17. Improve the availability of footpaths, and invest in cycling and other active transport infrastructure.[2 54 56 68 69 71] |

poverty.[24 46 47] The recognition that physical activity aids weight management may suggest that in Malawi, as elsewhere in sub-Saharan Africa,[23 48] Westernisation is leading to a shift from traditional weight preferences to a smaller body size. An alternative explanation may be the fact that our participants' exposure to health promotion information through involvement in the 2013-16 NCD survey encouraged them to provide views that would be socially acceptable from a public health perspective. Focus groups with people with suspected diabetes in Uganda, who may also have been exposed to health messaging, similarly revealed negative views about obesity, whereas focus groups with people without diabetes viewed excess body weight as a sign of success.[24]

Many of the barriers to being physically active reported in our study were similar to those found in previous studies in sub-Saharan Africa. These include negative social influences,[20–22 26] lack of time[20 21 23] and fear of injury.[20] In South Africa, issues around personal safety and crime emerged as major barriers to physical activity,

particularly for women.[21 22] However, our participants presented no such concerns. This may reflect the fact that Malawi currently experiences lower crime rates than most other sub-Saharan African countries,[49] but also that our participants were older, and perhaps felt less at risk than the younger participants in the other studies may have done.[50–52]

Importantly, almost all participants recognised the need to improve physical activity in Malawi. Table 2 summarises potential strategies to address barriers at different levels of the ecological model[35 36] for consideration in future interventions. The strategies reflect the barriers and opportunities identified by participants and previous evidence on the effectiveness of: embedding physical activity within daily routines;[53] highlighting the personal benefits/value of physical activity;[2 54] group-based exercising;[55] and promoting workplace-based physical activity.[56–58]

As Malawi continues to develop economically,[59 60] it is likely that (in part due to the societal expectations associated with personal wealth) requirements for occupational, transport and household physical activity will reduce,[61] meaning that recreational physical activity will become increasingly important to help people maintain healthy levels of physical activity.[56] The current study revealed that people in Malawi are already involved in a range of recreational physical activities. However, urban participants described engaging more in structured, facility-dependent exercising (although there was evidence that some groups felt excluded from gyms), while physical activity facilities were not mentioned by rural participants. These findings suggest that work needs to be done to ensure equality of opportunity in relation to physical activity in Malawi. This would include both investment in accessible physical activity facilities in rural areas, but also the promotion of less formal activities, such as dancing, walking and being active with family members,[21 62–65] in urban areas, to ensure that everyone can find a form of physical activity they enjoy.

This study had a number of strengths. First, its design (separate focus groups with men and women in urban and rural settings) allowed systematic comparison of the similarities and differences in men and women's experiences of, and attitudes towards, physical activity in urban and rural Malawi. In addition, the team approach to identifying and defining the analytical themes enhanced the robustness (validity) of our results.[66] However, it was not without limitations. Importantly, the participants had been told they were at high risk of developing diabetes following their participation in previous health research. All had been advised to attend their local health clinic or hospital, where they may have received information about the importance of physical activity for their health. They were also older, and some were dealing with other health conditions, so may have been more likely than the general adult population of Malawi to try to include different forms of physical activity in their daily lives. In addition, while focus groups were chosen to support the development of rich accounts of physical activity by enabling participants to build on each other's accounts, there is potential that participants may also adjust their own responses to conform to what they perceive as the views of other group members, thus limiting the richness of the data collected. However, the fieldworkers who conducted the focus groups did not take any contextual fieldnotes that would have supported a meaningful analysis of the extent to which the group setting influenced members' responses in this way.[67]

## CONCLUSION

This study provides detailed insights into the physical activity views and practices of men and women in urban and rural Malawi, which are important to inform future interventions to increase physical activity. The reported decline in routine physical activity is likely to continue as Malawi, like other countries in the sub-Saharan Africa region, undergoes social and economic transition, highlighting the need to promote recreational forms of physical activity. The findings suggest that in addition to information campaigns to raise awareness of the benefits of physical activity for all ages, the importance of finding forms of physical activity that people enjoy and the value of (and opportunities for) being physically active with other people, including children and grandchildren, should also be promoted. Finally, public investment in recreational facilities in rural and urban areas that are fully accessible for women, as well as men, and in active travel infrastructure may also be required to ensure that healthy levels of physical activity continue to be part of everyday life in Malawi.

**Acknowledgements** The authors would like to thank: Hazel Namadingo for her input while employed by MEIRU in designing the topic guide, conducting the focus groups in Lilongwe and quality assuring all transcripts; Cecelia Nyirenda for conducting the focus groups and undertaking the transcription and translation in Karonga; and Maisha Nyasulu for the transcription and translation in Lilongwe.

**Contributors** CMG is the guarantor of the conduct of this study. CMG, CB, JMRG and ACC designed and conducted the study and acquired the data. JB, CMG and CB conceived and designed the analyses. JB and CMG interpreted the data and contributed to drafting the manuscript. All authors contributed to revising the manuscript critically for important intellectual content and approved the version of the manuscript to be published.

**Funding** This study was funded by the Medical Research Council (MR/R019428/1).

**Disclaimer** The funder had no influence on the analysis plan, results presented or decision to publish.

**Competing interests** None declared.

**Patient and public involvement** Members of the public were involved in the design, or conduct, or reporting, or dissemination plans of this research. Refer to the Methods section for further details.

**Patient consent for publication** Not applicable.

**Ethics approval** This study involves human participants and was approved by the National Health Sciences Research Committee in Malawi (18/01/1951), and the University of Glasgow College of Medical, Veterinary and Life Sciences Ethics Committee in the UK (200170173). Participants gave informed consent to participate in the study before taking part.

**Provenance and peer review** Not commissioned; externally peer reviewed.

**ORCID iDs**
Jethro Banda http://orcid.org/0000-0001-8075-8564
Cindy M Gray http://orcid.org/0000-0002-4295-6110

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
