## [Reviewer comments · BMJ Open]

ARTICLE DETAILS

TITLE (PROVISIONAL)	A qualitative study of practices and attitudes towards physical activity among pre-diabetic men and women in urban and rural Malawi
AUTHORS	Banda, Jethro; Bunn, Christopher; Crampin, Amelia; Gill, Jason; Gray, Cindy

VERSION 1 – REVIEW

REVIEWER	Anna Chalkley Western Norway University of Applied Sciences Faculty of Teacher Education and Sport
REVIEW RETURNED	14-Dec-2021

GENERAL COMMENTS	Many thanks for the opportunity to review this manuscript. This study aimed to generate the in-depth understanding of physical activity practices and attitudes required to inform physical activity interventions in Malawi and use focus groups to collect data from men and women in rural and urban Malawi. The study generated some interesting findings which are important to help promote physical activity in Malawi and other low and middle income countries within sub-Saharan Africa. However, I question the analysis conducted and although there is something interesting and readable in the way the results are presented, it isn't clear to me how this links to the analysis and the themes which have been generated. Therefore, this makes me question its validity and I feel as if the analysis lacks analytical depth and interpretation of the data which is needed when exploring subjective accounts. Much more detail is needed to demonstrate how this aligns and I hope my feedback will be useful in strengthening this part of the paper.
--

REVIEWER	Svetlana Doubova Mexican Social Security Institute, Epidemiology and Health Services Research Unit, CMN Siglo XXI
REVIEW RETURNED	05-Jan-2022

GENERAL COMMENTS	The current version of the article presents several methodological concerns that should be addressed. First, there is a discrepancy between the study aims presented in the abstract and at the end of the introduction section. The aims of the study presented in both parts of the article should be similar to avoid misunderstanding. Also, the abstract of the article should be accurate, balanced and complete. The information on the results such as ("Thematic analysis identified 10 themes relating to physical activity practices and attitudes") as well as themes description should be presented in the result section instead of the method section.
---

	Second, the definition of the sample size and sampling strategy is not clear. The authors should specify how 37 study participants were selected from 50 pre-diabetic participants of the MRC-funded study? What was the response rate to participate in the study? How did the authors come up with the number of 4 focus groups? How can the authors be sure that they achieved sampling saturation with these four focus groups? Third, the choice of the focus groups as the data collection strategy is not clear; it will be helpful to explain the rationale behind using focus groups. Fourth, a central element of data analysis of the focus groups is an examination not only of the substantive content of the discussion but also the interaction between respondents themselves. Researchers who use focus groups and do not address the interaction between respondents will incompletely analyze the data. (Carey & Smith, 1994; Wainwright, 1994; Kitzinger, 1995; Johnson, 1996). The focus group analysis requires examination of three levels of units of analysis: (1) analysis at the individual level that examines the responses and behaviours without regard to the group context; (2) group-level analysis that includes interactional and sequential analysis; and (3) analysis of contextual aspects (group setting) that explains the relationship of the individual to the group. The focus group analysis should also include a comparison of the individual data with the group data. (Please see the reference: Carey MA, Smith MW. Capturing the Group Effect in Focus Groups: A Special Concern in Analysis. Qual Health Res 1994;4:123-127.) Currently, the article presents the individual-level analysis and the description of some contextual aspects; however, there is no clear group-level analysis that addresses the interaction among focus group participants. Fifth, it is also important to explain in the method section the authors' argument for performing thematic analysis as recommended by Braun and Clarke. Why is this method suited to the present study? At what level of abstraction did the authors aim in identifying themes? All themes that authors identified are purely descriptive and thus generally best classed as categories and not themes. Please, see the article "Confusing Categories and Themes" by J. Morse (2008). It is recommendable that the authors revise the level of analysis to reconsider whether they have sufficient data to report specific themes. Most themes are presented with a thin narrative and few quotes. Currently, it is difficult to appreciate the depth of the development of the analysis and the richness of the underlying data. For, instance, I did not find in the result section the narrative and quotes to support "social norms" and "sources of information" themes. Do the authors think they've adequately captured what the participants and data convey? I'd like to see more development here. Sixth, the authors should review the percentages presented in table one to ensure that for each variable the number of categories sum 100%; for instance, currently, the sum of the percentage for men and women who participated in the study is 100.1% instead of 100%. Seventh, it is important to include in the method and the discussion section the information on how the rigour of the study was established. Please, see the article "Critical Analysis of Strategies for Determining Rigor in Qualitative Inquiry" by J. Morse (2015). Eighth, to strengthen the discussion section, the researchers need to specify clearly the original contributions of their study, areas where findings corroborate existent literature, where the findings conflict, and limitations, as well as the relevance of the findings to the
--	---

	international readers. Also, the limitation section of the study requires the recognition of the censoring, conformity and group-thinking as the major limitations of the focus group technique.
--	--

REVIEWER	Samah Alageel King Saud University
REVIEW RETURNED	18-Jan-2022

GENERAL COMMENTS	Many thanks for the chance to read this very well written paper, below are minor comments for the authors to consider when writing the final draft to improve the overall paper. Overall: The fact that participants were selected because they were people with pre-diabetes need to be highlighted also in the title and the introduction. The authors discuss this in the limitations and discussion section, but making sure this is reflected on throughout the manuscript is needed. The factors that influence those at risk of diabetes are different than the factors that affect the general population. This population is expected to be older and dealing with other health conditions. Abstract: Report the themes in the results section. Introduction: Line 69: Do you have more recent statistics? Methods: Data analysis: was the ecological model applied to the types of PA only? or all themes? From reading the results it seems that it applies to all themes Results: I believe some of the sentences are actually discussion points. For example Page 13 Line 232 – 234 and Page 17 Line 319 – 322. There is an overlap between occupational and household activities, especially for Rural participants. Line 244: “to sell to make money” suggest rephrasing. The theme “factors influencing physical activity in Malawian adults” is great and very important I find the sub-themes very general. Suggest having more tailored sub-themes, which then can be linked to the ecological model, maybe in a figure or a table. Line 435: From the quote, I can also read that age is also a barrier feeling that gyms are not an inviting place for older participants. Discussion: Limitations: Suggest also discussing issues related to social desirability bias, the differences between those who agreed to be part of this study and those who did not participate and translation of data.
---

REVIEWER	Courtney Stevens Dartmouth-Hitchcock Medical Center, Psychiatry
REVIEW RETURNED	21-Jan-2022

GENERAL COMMENTS	The aim of the study is stated differently in the abstract and the introduction sections. The framing of the study aim in the introduction is a run-on sentence and difficult to follow. There is no description of research ethics (informed consent, etc.) in the Method.
---

VERSION 1 – AUTHOR RESPONSE

Reviewer 1	
Title: Given the aim of the paper, I would recommend adding Pre-diabetic into the title to read 'A qualitative study of practices and attitudes towards physical activity among men and women in urban and rural Malawi'	Thank you for this suggestion; we agree and have amended the title to 'A qualitative study of practices and attitudes towards physical activity among pre-diabetic men and women in urban and rural Malawi' p1 In1-2.
Introduction: I do not feel as if it is clear as to why Malawi specifically has been chosen as the geographical area of interest for the study. The authors refer to inactivity levels and the implications of this on NCDs but perhaps a comparison of Malawi to other areas within sub Saharan Africa would be useful to make a clear rationale?	Malawi was chosen simply because it is the country where MEIRU is based, rather than because of its levels of inactivity and NCDs in relation to other Sub-Saharan African countries. As there has been little previous research (particularly qualitative) on physical activity in Malawi (as mentioned in the Introduction), this paper makes an important contribution to address this evidence gap. This is also highlighted in the Discussion on p24 In518-521.
Introduction: Check the use of upper and lower case used for Sub-Saharan Africa, currently there is a mixture throughout the paper (eg. Line 66 vs line 77 in the introduction).	Thank you for this comment, we have amended the use of case to Sub-Saharan Africa throughout.
Methods: I think it would be useful to add in a design section within the methods which details why a qualitative approach was chosen and the appropriateness of focus groups to capture accounts of physical activity and specifically attitudes to physical activity. I feel as this would also help to address some of the missing detail within the SRQR checklist. Simply engaging in focus group interviews does not make a qualitative study (it was simply a qualitative method of data generation). To ensure methodological coherence the authors should explicitly state how their philosophical worldviews informed their study design (e.g., qualitative description, phenomenology, etc.).	We agree and have added a study design section to the Methods, p7 In125-133, justifying our approach and pragmatist philosophy,
Setting/Participants: There is currently no information relating to how participants were recruited for the focus groups specifically. Was this purposive based on location (to ensure a balance of urban and rural) for example? More detail is needed. Specifically, how did you manage the process of recruitment and analysis to test for saturation? Too often this detail is omitted, but reporting it could be very useful to other researchers (since this is not a straightforward thing to do). e.g. Did	We have added this into a revised section, now entitled 'Setting and Recruitment', p8 In135-144 We did not deploy the concept of saturation in our work. While we appreciate that this concept is important to those using grounded theory and has found use in other approaches to qualitative research, as Saunders et al argue (DOI: 10.1007/s11135-017-0574-8), we note that it not part of the thematic analysis approach we used. https://www.ncbi.nlm.nih.gov/pmc/articles/PMC5993836/

you recruit groups one by one, analyse as you went, then decide whether to continue (if this were the case, how did you ensure a mix of urban and rural participants were recruited)? Or did you recruit the whole sample, then collect data only until saturation (and inform the others they would no longer be needed)? Or did you recruit the whole sample and collect data in one phase, and it happened by chance that saturation was reached when you had finished? Please make these processes transparent.	
Data collection and management: Can the authors add in some detail regarding the development of the interview guide e.g. was it theory informed? Literature informed?	Development of the topic guide was informed by our pragmatist approach. This detail has been added to p9 In160.
Data collection and management: Can the authors comment on how contextual meaning of the data was retained during the translation of the interview guide and transcription. This information is important for demonstrating trustworthiness of the data and rigour (e.g. is JB a native speaker of Chichewa and Chitumbuka and if not, how might this have influenced the quality assurance?	Translation of the topic guide was done by fluent Chichewa and Chitumbuka speakers, and quality assurance was carried out by native or fluent speakers of these languages. This detail has been added to p9 In167-176.
Data analysis: would question the use of reflexive TA (as described by Braun and Clarke), can the authors comment on the appropriateness of this method for comparing and contrasting accounts of physical activity, as per the aim (line 117).	The comparison of accounts was undertaken using detailed notetaking and an adapted OSOP. This detail has been added to p11 In203 and In212-213.
Data analysis: [ ] Assuming that reflexive thematic analysis is fitting, more detail is needed in terms of how the analysis took a reflective approach e.g. did (and if so how) the coding framework change when applied to the transcripts (i.e. how was it an iterative process?) and what this looked	More detail of the analysis process is provided (p10 In178-205), which describes the reflective and iterative approach used (e.g., p10 In193-205). We have amended Table S1 in the Supplementary Materials to include example extracts for each of the themes.

like within the data. A table detailing the theme, description and an example quotation might be useful to demonstrate the link.	
Data analysis: □ The analysis appears to lack some analytical depth and offers very surface level descriptions.	More detail of the analysis process is provided (p10 ln178-205).
Results: From reading the results it seems to me that there are some interesting findings about conflicting priorities and activity done out of necessity rather than choice. I was also surprised not to see anything about equity and accessibility of physical activity opportunities.	We have added an extract about physical activity and necessity p18-19 ln373-379. There is a quote about women feeling excluded from some gyms because of the mainly male clientele on p22 ln460-464, and we have added consideration about the fact that many formal physical activity opportunities did not appear to be available to rural participants and some urban groups in the discussion p28 ln577-585.
Results: The results section is fairly long and I wondered if the authors had considered presenting the 10 themes within the results and using the ecological model as a framework to structure the discussion and offer some interpretation? This would help reduce some of the repetitiveness between the results and discussion.	Thank you for this suggestion, we agree and have tried our best to follow it p17-23 337-489, p24 ln511-515, and Table 2, p27.
Results: I would question whether some of the factors identified are represented as the right level of the ecological model e.g. P 17 line 355 I do not think multiple competing pressures are an individual level influence, neither family responsibilities on line 367 p 19	We have changed the structure of the results section and have included summary of our analysis in Supplementary materials Table S3 in which we have represented family responsibilities and pressure to make money at the interpersonal level.
Results: Wherever intensity of the activity is reported I think it would be useful to clarify that this is as perceived by the participant e.g. line 224 p 12 and 13	The intensity of activity is not as perceived as the participant, rather it is a classification applied by the researchers.
Results: Line 232 – 234 I feel as if this is more interpretive and should be used in the discussion rather than the results section.	We agree and have removed all interpretation (this and elsewhere) from the results.
Results: P 20 line 392 I am interested in the finding that participants described themselves as lazy, is there anything in the data which would allow the authors to	Lazy appears to have negative connotations and to be associated with risk of disease and weakness across participants' accounts. This point has been added to p20 ln414-417.

elaborate a little more on this	
Discussion: Line 515 I'd suggest referring to the WHO guidelines which would be more relevant than the UK PA guidelines where strength was also present in the previous 2011 iteration.	We are using the UK Guidance to make the point that the importance of strength exercising (and the fact that uptake is low) is gaining increasing prominence but agree that WHO guidance should also be referenced due to their relevance to Malawi (p25 ln533-536).
Discussion: Line 525 The desire to lose weight was not mentioned in the results section (apologies if I missed this) but new results should not be presented in the discussion.	We agree that new results should not be presented in the discussion – weight loss was mentioned in relation to physical activity and health in the results – now p18 ln353-360.
Discussion: Table 2 – really useful addition and pleased to see some practical (and evidence based) recommendations	Thank you.
Reviewer 2	
First, there is a discrepancy between the study aims presented in the abstract and at the end of the introduction section. The aims of the study presented in both parts of the article should be similar to avoid misunderstanding.	We have amended the aim in the introduction to reflect more closely the aim in the abstract on p7 ln118-122.
Also, the abstract of the article should be accurate, balanced and complete. The information on the results such as (“Thematic analysis identified 10 themes relating to physical activity practices and attitudes”) as well as themes description should be presented in the result section instead of the method section.	We have considered this suggestion, but our normal practice is to present the themes in the method, rather than the results (e.g., doi:10.1093/geront/gny020), and we would prefer to maintain this style in the current manuscript.
Second, the definition of the sample size and sampling strategy is not clear. The authors should specify how 37 study participants were selected from 50 pre-diabetic participants of the MRC-funded study? What was the response rate to participate in the study? How did the authors come up with the number of 4 focus groups? How can the authors be sure that they achieved sampling saturation with these four focus groups?	Detail of the sampling (purposive to achieve variation in age), and 100% response rate on p8 ln141-144. We conducted four focus groups to allow men and women to contribute their views separately in each location (urban/rural). We did not deploy the concept of saturation. While we appreciate that this concept is important to those using grounded theory and has found use in other approaches to qualitative research, as Saunders et al argue (DOI: 10.1007/s11135-017-0574-8), we note that it not part of the thematic analysis approach we used.
Third, the choice of the focus groups as the data collection strategy is not clear; it will be helpful to explain the rationale behind using focus groups.	We have described our rationale for using focus groups in an additional study design section on p7 ln124-133.
Fourth, a central element of data analysis of the focus groups is an examination not only of the substantive content of the discussion but also the interaction between respondents themselves. Researchers who use focus groups and do not address the interaction between respondents will incompletely analyze the data. (Carey & Smith, 1994; Wainwright, 1994; Kitzinger,	Thank you for this important comment. We have gone back to our data to examine the interactions and discourse between participants, and have highlighted these, where appropriate, in the results (e.g. p18 ln354-355 and 366-372, p19 ln381-382).

1995; Johnson, 1996). The focus group analysis requires examination of three levels of units of analysis: (1) analysis at the individual level that examines the responses and behaviours without regard to the group context; (2) group-level analysis that includes interactional and sequential analysis; and (3) analysis of contextual aspects (group setting) that explains the relationship of the individual to the group. The focus group analysis should also include a comparison of the individual data with the group data. (Please see the reference: Carey MA, Smith MW. Capturing the Group Effect in Focus Groups: A Special Concern in Analysis. Qual Health Res 1994;4:123-127.) Currently, the article presents the individual-level analysis and the description of some contextual aspects; however, there is no clear group-level analysis that addresses the interaction among focus group participants.	
Fifth, it is also important to explain in the method section the authors' argument for performing thematic analysis as recommended by Braun and Clarke. Why is this method suited to the present study? At what level of abstraction did the authors aim in identifying themes? All themes that authors identified are purely descriptive and thus generally best classed as categories and not themes. Please, see the article "Confusing Categories and Themes" by J. Morse (2008). It is recommendable that the authors revise the level of analysis to reconsider whether they have sufficient data to report specific themes. Most themes are presented with a thin narrative and few quotes. Currently, it is difficult to appreciate the depth of the development of the analysis and the richness of the underlying data. For, instance, I did not find in the result section the narrative and quotes to support "social norms" and "sources of information" themes. Do the authors think they've adequately captured what the participants and data convey? I'd like to see more development here.	Thank you for this critique: We have provided more detail of how we performed our thematic analysis in the data analysis section on p9-11 In177-216, and have provided two tables to support our analysis in Supplementary materials (Tables S2 and S3). In addition, we have revised our results section, following the suggestion from Reviewer 1, to present the results according to the themes. We hope these steps adequately address this point.
Sixth, the authors should review the percentages presented in table one to ensure that for each variable the number of categories sum 100%; for instance, currently, the sum of the percentage for men and women who participated in the study is 100.1% instead of 100%.	Thank you for pointing this out. We have rounded the percentages appropriately where possible. However, the six urban males are spread equally into three age groups (33.3% in each), which is not possible to round to 100%. Table 1 p12.
Seventh, it is important to include in the method and the discussion section the information on how the rigour of the study	We have provided more detail of the analysis process followed in the data analysis section (p9-11 In178-216), which describes how three of the authors worked

was established. Please, see the article "Critical Analysis of Strategies for Determining Rigor in Qualitative Inquiry" by J. Morse (2015).	together to ensure rigour. In addition, we have referenced this paper (thank you for pointing us to it) as support for the rigor of this approach in the Discussion on p28 ln589-590.
Eighth, to strengthen the discussion section, the researchers need to specify clearly the original contributions of their study, areas where findings corroborate existent literature, where the findings conflict, and limitations, as well as the relevance of the findings to the international readers. Also, the limitation section of the study requires the recognition of the censoring, conformity and group-thinking as the major limitations of the focus group technique.	Thank you for this important comment. We have highlighted the original contributions of this study (in providing the first in-depth qualitative exploration of the factors underpinning urban/rural and gender differences in physical activity among Malawian adults) on p24-25 ln517-523, corroboration with previous studies (p26 ln551-555 and 556-558) and conflicts with previous studies (e.g., p25-26 ln542-545, p26 ln558-563). We have also added a sentence to recognise the potential for conformity in the focus group discussions to the Discussion on p29 ln597-601. Finally, given the similarity of the Malawian context to other Sub-Saharan African countries, the findings will be of relevance across the region. We have highlighted this similarity in the first line of the Abstract on p2 ln20, and in the Conclusion on p29 ln605-608.
Reviewer 3	
Overall: The fact that participants were selected because they were people with pre-diabetes need to be highlighted also in the title and the introduction. The authors discuss this in the limitations and discussion section, but making sure this is reflected on throughout the manuscript is needed. The factors that influence those at risk of diabetes are different than the factors that affect the general population. This population is expected to be older and dealing with other health conditions.	We have amended the title (p1 ln2) and have highlighted the fact that the participants had prediabetes in the introduction (p6 ln120) and elsewhere in the manuscript (e.g., p7 ln126), we have also included specific reference to how age and other health conditions might have influenced their views on p29 ln595-597, as we agree these are important.
Abstract: Report the themes in the results section.	See response to Reviewer 2 on the same point
Introduction: Line 69: Do you have more recent statistics?	We have reported the most recent World Health Organization Country profiles and have amended the text on p5 ln68-69 to make this clear. Other Malawian survey studies (referenced in the introduction) also date from a few years ago, and we have not found more up-to-date studies. We would be happy to include any others that the reviewer is aware of.
Methods: Data analysis: was the ecological model applied to the types of PA only? or all themes? From reading the results it seems that it applies to all themes	We have amended the data analysis section to make it clear that the application of the ecological model also covered influences on physical activity (p10-11 ln206-210) and have provided Table S2 in Supplementary materials to provide detail of the results of this analysis.
Results: I believe some of the sentences are actually discussion points. For example	We agree and have removed these from the results section, which also helps reduce the repetition in the manuscript.

Page 13 Line 232 – 234 and Page 17 Line 319 – 322.	
There is an overlap between occupational and household activities, especially for Rural participants.	We agree and have highlighted this on p14 ln274-275.
Line 244: “to sell to make money” suggest rephrasing.	Having considered other alternatives, we think this phrasing works best, but are happy to consider any specific suggestions that we may not have thought of.
The theme “factors influencing physical activity in Malawian adults” is great and very important I find the sub-themes very general. Suggest having more tailored sub-themes, which then can be linked to the ecological model, maybe in a figure or a table.	We have followed the suggestion from Reviewer 1 to restructure our Results section according to our broad analytic themes (p17-23 337-489). We have also provided a Table S3 in Supplementary materials which links the sub-themes to the levels and domains of the ecological model.
Line 435: From the quote, I can also read that age is also a barrier feeling that gyms are not an inviting place for older participants.	We agree and have added this dimension to p22 ln460-462.
Discussion: Limitations: Suggest also discussing issues related to social desirability bias, the differences between those who agreed to be part of this study and those who did not participate and translation of data.	We do not consider social desirability bias to be a major issue in this study, as all participants in the main study who were eligible to take part in a focus groups agreed to do so. This detail is now provided on p8 ln141-144. In addition, the analysis was led by JB who is a native Chichewa speaker and fluent in Chitumbuka (as well as in English), and in cases where there was doubt over interpretations during the translation into English (which was necessary to allow the UK team to fully engage in the analysis), JB was able to consult the original data.
Reviewer 4	
The aim of the study is stated differently in the abstract and the introduction sections.	We have amended the aim in the Introduction to reflect more closely the aim in the abstract – p7 ln118-122
The framing of the study aim in the introduction is a run-on sentence and difficult to follow.	We have amended the aim in the introduction to align it more with the abstract, and in doing so have also tried to make it more readable – p7 ln116-122. We hope we have succeeded.
There is no description of research ethics (informed consent, etc.) in the Method.	As per the journal style, the ethics statement is provided at the end of the manuscript on p30-31 ln636-639. Consent is also mentioned on p9 ln158-159. We have also added information about informed consent to the Setting and recruitment section on p8 ln141-143.

VERSION 2 – REVIEW

REVIEWER	Anna Chalkley Western Norway University of Applied Sciences Faculty of Teacher Education and Sport
REVIEW RETURNED	09-Aug-2022
GENERAL COMMENTS	Many thanks for the opportunity to review this revised manuscript which has much improved following the revisions made in response to previous comments. I have two points which should be addressed before recommending for acceptance: Firstly, given the additional detail included in the methods section it is my belief that the authors may have confused Braun and Clarke's

	approach to reflexive thematic analysis with codebook approaches to TA. It might be useful for the authors to review Braun et al 2019 (Braun, V., Clarke, V.: Reflecting on reflexive thematic analysis. Qual. Res. Sport Exerc. Health 11(4), 589–597 (2019). https://doi.org/10.1080/2159676X.2019.1628806) in which they demarcate the position of RTA among other forms of TA. Codebook thematic analysis typically involves developing codes before/at the start of the analysis process whereas reflexive thematic analysis usually involves creating these at a later stage of the analysis. I believe it is a legitimate approach for their piece of work but this is an important distinction which would need to be made before the manuscript can be accepted. Secondly, I think the application of the ecological model to frame the results works well and this section is much clearer but think it is still important to report the 10 themes identified via the analysis within the main results section at the beginning (although further detail is given in supplementary material). The authors could have coded using the ecological model themes, but didn't, so it is important to describe the added value that the analysis gave. As an introduction to this section, perhaps move the text (lines 208-212 to the text at line 234 to introduce the themes within the context of the model?
--	--

REVIEWER	Svetlana Doubova Mexican Social Security Institute, Epidemiology and Health Services Research Unit, CMN Siglo XXI
REVIEW RETURNED	16-Aug-2022

GENERAL COMMENTS	I find the revised version slightly improved, yet the authors addressed multiple reviewers' concerns incompletely. For instance, regarding the suggestion to present the themes description in the study results section instead of the method section, the authors answered that their normal practice is to present the themes in the methods. To support this response, the authors provided the reference (doi:10.1093/geront/gny020) that does not support the above-mentioned authors' justification for maintaining the description of the themes as part of the methods, as in the above-mentioned article, the themes are reported in the results section of the abstract. Another example of the authors addressing reviewers' concerns incompletely is the response regarding the discrepancy between the study objective in the abstract and at the end of the introduction section that remains different in the revised version of the article. In the abstract the authors specified that "This study aimed to generate the in-depth understanding of physical activity practices and attitudes required to inform physical activity interventions in Malawi." However, at the end of the introduction section the authors specified that "the aim of this qualitative study was to compare and contrast accounts of practices and attitudes towards physical activity among Malawian men and women in urban and rural settings." Another example of the authors addressing reviewers' concerns incompletely is the response regarding the study sample size and sampling. The authors did not answer why they chose 32 participants from 50 prediabetic patients identified through a 2013-2016 NCD survey. On the one side, the authors said that they did not apply the saturation technique to define the study sample size as, according to the authors, this is not a part of the thematic analysis approach they used (this statement lacks clear justification). On the other side, the authors said that they interviewed 32 out of 50 available participants with a 100% response rate. Yet, it is still
---

	unclear why only 32 out of 50 prediabetic patients participated. How did the authors come up with a sample size of 32? Regarding the study analysis, the authors specified that they performed a reflexive thematic approach. Yet, the description of the study analysis is more close to the "coding reliability thematic analysis" approach, while lacking essential characteristics of the reflexive thematic approach. According to Braun & Clarke "reflexive thematic approach uses open, fluid, organic, and recursive coding practices. In reflexive thematic analysis, codes are never finally fixed. They can evolve, expand, contract, be renamed, split into several codes, collapse together with other codes, and even be abandoned. In reflexive thematic analysis, coding becomes more interpretive and conceptual across an analysis, moving beyond the surface and explicit meaning to interrogate implicit (latent) meaning. Such developments and refinements reflect the researcher's deepening engagement with their data and their evolving, situated, reflexive interpretation of them. They also demonstrate a key point for reflexive thematic analysis: codes are conceptual tools in the developing analysis and should not be reified into ontologically real things, while themes are conceptualized as patterns of shared meaning underpinned or united by a core concept". ('central organizing concept')" (Please see, Braun, V., Clarke, V., Terry, G & Hayfield N. (2018). Thematic analysis. Liamputtong, P. (Ed.), Handbook of research methods in health and social sciences (pp. 843-860). Singapore: Springer and Braun, V., Clarke, V. (2019) Reflecting on reflexive thematic analysis. Qualitative Research in Sport, Exercise and Health 11;589-597.) Unfortunately, I could not clearly identify the features of the reflexive thematic approach in the description of the study analysis and study results. Despite the response of the authors that they have gone back to the data to examine the interactions and discourse between focus group participants and have highlighted these, where appropriate (e.g. p18 ln354-355 and 366-372, p19 ln381-382), I did not find an explicit description of the focus group analysis in the method and the results sections that addresses examination of three levels: (1) analysis at the individual level that examines the responses and behaviors without regard to the group context; (3) analysis of contextual aspects (group setting) that explains the relationship of the individual to the group; and also includes a comparison of the individual data with the group data. (Please see the reference: Carey MA, Smith MW. Capturing the Group Effect in Focus Groups: A Special Concern in Analysis. Qual Health Res 1994;4:123-127.)
--	--

VERSION 2 – AUTHOR RESPONSE

Reviewer 1	
Firstly, given the additional detail included in the methods section it is my belief that the authors may have confused Braun and Clarke's approach to reflexive thematic analysis with codebook approaches to TA. It might be useful for the authors to review in which they demarcate the position of RTA among other forms of TA. Codebook thematic analysis typically involves developing codes before/at the start	After careful consideration and reading of Braun, V., Clarke, V., Terry, G & Hayfield N. (2018). Thematic analysis. Liamputtong, P. (Ed.), Handbook of research methods in health and social sciences (pp. 843-860). Singapore: Springer and Braun, V., Clarke, V. (2019) Reflecting on reflexive thematic analysis. Qualitative Research in Sport,

of the analysis process whereas reflexive thematic analysis usually involves creating these at a later stage of the analysis. I believe it is a legitimate approach for their piece of work but this is an important distinction which would need to be made before the manuscript can be accepted.	Exercise and Health 11;589-597 we agree that our approach aligns more closely with the codebook thematic approach and have amended the manuscript accordingly (e.g. In 29, 180, 187).
Secondly, I think the application of the ecological model to frame the results works well and this section is much clearer but think it is still important to report the 10 themes identified via the analysis within the main results section at the beginning (although further detail is given in supplementary material). The authors could have coded using the ecological model themes, but didn't, so it is important to describe the added value that the analysis gave. As an introduction to this section, perhaps move the text (lines 208-2012 to the text at line 234 to introduce the themes within the context of the model?	We have moved the description of the themes to the results section (lines 223-232). We have also moved the detailed description of the ecological model to the results (lines 238-241)
Reviewer 2	
Regarding the suggestion to present the themes description in the study results section instead of the method section, the authors answered that their normal practice is to present the themes in the methods. To support this response, the authors provided the reference (doi:10.1093/geront/gny020) that does not support the above-mentioned authors' justification for maintaining the description of the themes as part of the methods, as in the above-mentioned article, the themes are reported in the results section of the abstract.	We believe the reviewer may have mis read the paper reference, which clearly reports the themes identified in the analysis in the method section as follows: 'An initial coding frame with eight broad themes was constructed, guided by the research questions and emergent ideas. The eight broad themes were as follows: Description of sitting/non-sitting activities; Perceptions of sitting/non-sitting; Health and well-being; Function; Other people; Hobbies and interests; Changing/not changing sitting behavior; and Changes in sitting over time.' Nevertheless, given the fact that Reviewer 1 would also prefer the description of the themes, we have now moved this to the Results section (lines 223-232).
The response regarding the discrepancy between the study objective in the abstract and at the end of the introduction section that remains different in the revised version of the article. In the abstract the authors specified that "This study aimed to generate the in-depth understanding of physical activity practices and attitudes required to inform physical	We have rewritten the abstract to conform to BMJ Open formatting. The study objective in the abstract now reads "this study aimed to compare and contrast accounts of practices and attitudes towards physical activity among Malawian men and women (previously identified as having pre-diabetes)

activity interventions in Malawi.” However, at the end of the introduction section the authors specified that “the aim of this qualitative study was to compare and contrast accounts of practices and attitudes towards physical activity among Malawian men and women in urban and rural settings.”	in urban and rural settings.” (lines 22-24).
The authors did not answer why they chose 32 participants from 50 prediabetic patients identified through a 2013-2016 NCD survey. On the one side, the authors said that they did not apply the saturation technique to define the study sample size as, according to the authors, this is not a part of the thematic analysis approach they used (this statement lacks clear justification). On the other side, the authors said that they interviewed 32 out of 50 available participants with a 100% response rate. Yet, it is still unclear why only 32 out of 50 prediabetic patients participated. How did the authors come up with a sample size of 32?	The reference to a 100% response rate in the previous response to reviewers related to the fact that we had a pool of 50 participants from the larger study who agreed to be interviewed (i.e. there were no refusals to taking part in a future interview if invited during the informed consent process for the larger study), rather than the sampling (see lines 143-145). The fieldworkers were instructed to invite around 10 participants for each gender and each setting to ensure a variation in age and enough participants (considered locally to be at least four) for each focus group. The sample of 32 reflects those who showed up for each focus group. In relation to saturation, as discussed by Braun & Clarke DOI: 10.1080/2159676X.2019.1704846, our sample size was determined by the pragmatic constraints of the local context (p. 211) as well that fact that saturation was not a requirement of our analysis approach.
Regarding the study analysis, the authors specified that they performed a reflexive thematic approach. Yet, the description of the study analysis is more close to the "coding reliability thematic analysis" approach, while lacking essential characteristics of the reflexive thematic approach.	After careful consideration and reading of Braun, V., Clarke, V., Terry, G & Hayfield N. (2018). Thematic analysis. Liamputtong, P. (Ed.), Handbook of research methods in health and social sciences (pp. 843-860). Singapore: Springer and Braun, V., Clarke, V. (2019) Reflecting on reflexive thematic analysis. Qualitative Research in Sport, Exercise and Health 11;589-597 we agree that our approach aligns more closely with the codebook thematic approach and have amended the manuscript accordingly (e.g. lines 29, 180, 187).
Despite the response of the authors that they have gone back to the data to examine the interactions and discourse between focus group participants and have highlighted these, where appropriate (e.g. p18 ln354-	In order to compare and contrast accounts of practices and attitudes towards physical activity among Malawian men and women in urban and rural settings, our analysis of the

355 and 366-372, p19 In381-382), I did not find an explicit description of the focus group analysis in the method and the results sections that addresses examination of three levels: (1) analysis at the individual level that examines the responses and behaviors without regard to the group context; (3) analysis of contextual aspects (group setting) that explains the relationship of the individual to the group; and also includes a comparison of the individual data with the group data.	focus group transcripts does focus on individual responses and group interactions. It did not examine the influence of the group setting and we did not have access to any contextual fieldnotes that would have supported such an analysis. We have acknowledged this limitation in Strengths and Limitations (lines 58-60) and Discussion sections (lines 603-606).
--	--

VERSION 3 – REVIEW

REVIEWER	Anna Chalkley Western Norway University of Applied Sciences Faculty of Teacher Education and Sport
REVIEW RETURNED	04-Nov-2022c
GENERAL COMMENTS	Thank you for the revised manuscript. I am satisfied that the changes made address the points raised in the review and have strengthened the paper for publication.